# Limitation of adipose tissue by the number of embryonic progenitor cells

Kristina Hedbacker[1,2], Yi-Hsueh Lu[1], Olof Dallner[1], Zhiying Li[1], Gulya Fayzikhodjaeva[1,2], Kıvanç Birsoy[1,3], Chiayun Han[4], Chingwen Yang[4], Jeffrey M Friedman[1,2]*

[1]Laboratory of Molecular Genetics, The Rockefeller University, New York, United States; [2]Howard Hughes Medical Institute, The Rockefeller University, New York, United States; [3]Laboratory of Metabolic Regulation and Genetics, The Rockefeller University, New York, United States; [4]Gene Targeting Resource Center, The Rockefeller University, New York, United States

**Abstract** Adipogenesis in adulthood replaces fat cells that turn over and can contribute to the development of obesity. However, the proliferative potential of adipocyte progenitors in vivo is unknown (Faust et al., 1976; Faust et al., 1977; Hirsch and Han, 1969; Johnson and Hirsch, 1972). We addressed this by injecting labeled wild-type embryonic stem cells into blastocysts derived from lipodystrophic A-ZIP transgenic mice, which have a genetic block in adipogenesis. In the resulting chimeric animals, wild-type ES cells are the only source of mature adipocytes. We found that when chimeric animals were fed a high-fat-diet, animals with low levels of chimerism showed a significantly lower adipose tissue mass than animals with high levels of chimerism. The difference in adipose tissue mass was attributed to variability in the amount of subcutaneous adipose tissue as the amount of visceral fat was independent of the level of chimerism. Our findings thus suggest that proliferative potential of adipocyte precursors is limited and can restrain the development of obesity.

*For correspondence:
friedj@mail.rockefeller.edu

Competing interests: The authors declare that no competing interests exist.

## Introduction

Obesity is associated with a set of metabolic abnormalities, collectively known as the metabolic syndrome, which includes insulin resistance and diabetes, dyslipidemia, hypertension and an increased risk of cardiac disease (*Grundy, 2004*). These same abnormalities also develop in lipodystrophy, a condition resulting from a reduced adipose tissue mass. The fact that both increased and decreased adipose tissue mass can cause metabolic disease has suggested that when the storage capacity of adipose tissue is exceeded, the result is 'lipid overflow', the deposition of excess lipid in peripheral tissues, leading to lipotoxicity, insulin resistance, and metabolic disease (*Huang-Doran et al., 2010*). This hypothesis suggests that the mechanisms that control adipose tissue development and the total amount (mass) of adipose tissue in vivo will be important for the pathogenesis of metabolic disease.

While the program controlling adipose tissue differentiation has been extensively studied in vitro (*Rosen and Spiegelman, 2000*), less is known about the cellular mechanisms controlling adipose tissue formation in vivo or the factors that regulate its size (mass) (*Rodeheffer et al., 2008*; *Lee et al., 2012*). In adults, homeostasis of adipose tissue mass is controlled by a negative feedback loop comprised of the hormone leptin and a set of neural targets that regulate food intake and metabolism (*Friedman and Halaas, 1998*). Surgical lipectomy leads to restoration of the removed fat presumably as a result of reduced levels of leptin (*Hernandez et al., 2011*; *Reyne et al., 1983*; *Knittle and Hirsch, 1968*). Reduced leptin signaling also leads to obesity in ob mice that lack leptin, and in animals fed a high fat diet (HFD) which causes leptin resistance (*Friedman and Halaas, 1998*).

In normal lean animals, the total number and size of adipocytes remains largely unchanged after adolescence (*Knittle and Hirsch, 1968*; *Greenwood and Hirsch, 1974*; *Greenwood et al., 1979*) with replacement of adipocytes that have turned over by newly formed adipocytes (*Stiles et al., 1975*; *Hemmeryckx et al., 2010*). The development of obesity in adults on a high fat diet, is a result of both adipocyte hypertrophy, increased cell size, and hyperplasia, the formation of new fat cells (*Spalding et al., 2008*; *Wang et al., 2013*). Together, these observations suggest that homeostatic control of adipose tissue mass requires the coordinated regulation of adipocyte growth and proliferation. The regulation of adipogenesis is thus of great importance because as mentioned, several lines of evidence have suggested that metabolic disease can develop when the capacity of adipose tissue to store lipid becomes limited, leading to its deposition in peripheral tissues (*Huang-Doran et al., 2010*; *Unger, 2003*; *Medina-Gomez et al., 2007*). However, the mechanisms responsible for the increase in fat tissue in adults are not well understood and the possibility that the proliferative potential of adipocyte precursors can be limited has not been directly studied.

We thus set out to test whether adipose tissue mass in adults is correlated with the degree of chimerism in lipodystrophic animals derived from AZIP blastocysts into which wild-type ES cells were injected (*Moitra et al., 1998*). This approach is analogous to one that has previously been used to assess whether the number of progenitor cells could be limiting for pancreas and liver development. In that study, the numbers of embryonic pancreatic and liver progenitors were inferred by determining the degree of chimerism in adult animals using immunohistochemistry (*Stanger et al., 2007*). In studies of pancreas, wild-type ES cells were injected into blastocysts from apancreatic *Pdx1*-deficient mice and the authors concluded that a smaller number of pancreatic progenitor cells was associated with a diminished pancreas size. In contrast, this same paper concluded that liver size is not limited by the number of progenitor cells (*Stanger et al., 2007*). In our study, we used a similar ES cell complementation strategy to determine whether the level of chimerism in adult animals is correlated with the number of adipocytes and total adipose tissue mass and whether a decreased storage capacity of adipose tissue would be associated with metabolic abnormalities.

## Results

### Restoration of adipose tissue mass after AZIP-Blastocyst complementation

Wild-type:AZIP-chimeras were generated by injecting CAG-hTubYFP mouse B6-*Tyr*$^c$ ES-cells (ES-cells hereafter) into 3.5 day blastocysts generated by breeding wild-type (WT) FVB females to AZIP (FVB) transgenic males (*Figure 1A*). (Note, FVB mice do not become as obese as C57Bl/6J mice when fed a high fat diet so in subsequent experiments we also studied B6 WT:AZIP chimeras, see below). The A-ZIP transgenic animals carry a dominant negative form of cEBPα leading to a block in adipogenesis resulting in severe congenital lipodystrophy (*Moitra et al., 1998*). We reasoned that because the WT ES-cells did not express the cEBPα transgene, the injected WT ES-cells should reconstitute the adipose tissue and that, if so, all adipocytes in the chimeric animals would be labeled by YFP. Cells derived from ES-cells were identified by expression of YFP using immunohistochemistry or by PCR of DNA from adipocyte DNA using YFP primers.

All of the adipocytes from WT:AZIP-chimeric mice expressed *YFP,* assessed using IHC, confirming that the WT ES-cells were the only source of mature adipocytes (*Figure 1B*). As expected, other tissues, including the adipose tissue stromal fraction, showed both *YFP*-positive and negative cells. The extent of chimerism was quantified in a manner similar that used in a prior publication by calculating the ratio between the amount YFP-DNA (found only in cells derived from the injected WT ES-cells) and that of control genes, FABP4 or RPL23, which are similarly expressed in all cells (*Stanger et al., 2007*). Note, one cannot use adipose tissue from WT:AZIP chimeras to calculate the extent of chimerism since adipocytes are exclusively derived from the ES cells. Half of the blastocysts in each experiment do not carry the A-ZIP transgene and we used these animals derived from WT embryos receiving WT ES cells, referred to as WT:WT chimeras, as a further control. These animals are similarly chimeric in all tissues including adipose tissue and characterization of these animals confirmed that the extent of chimerism in blood cells was well correlated with the extent of chimerism in adipose tissue (data not shown).

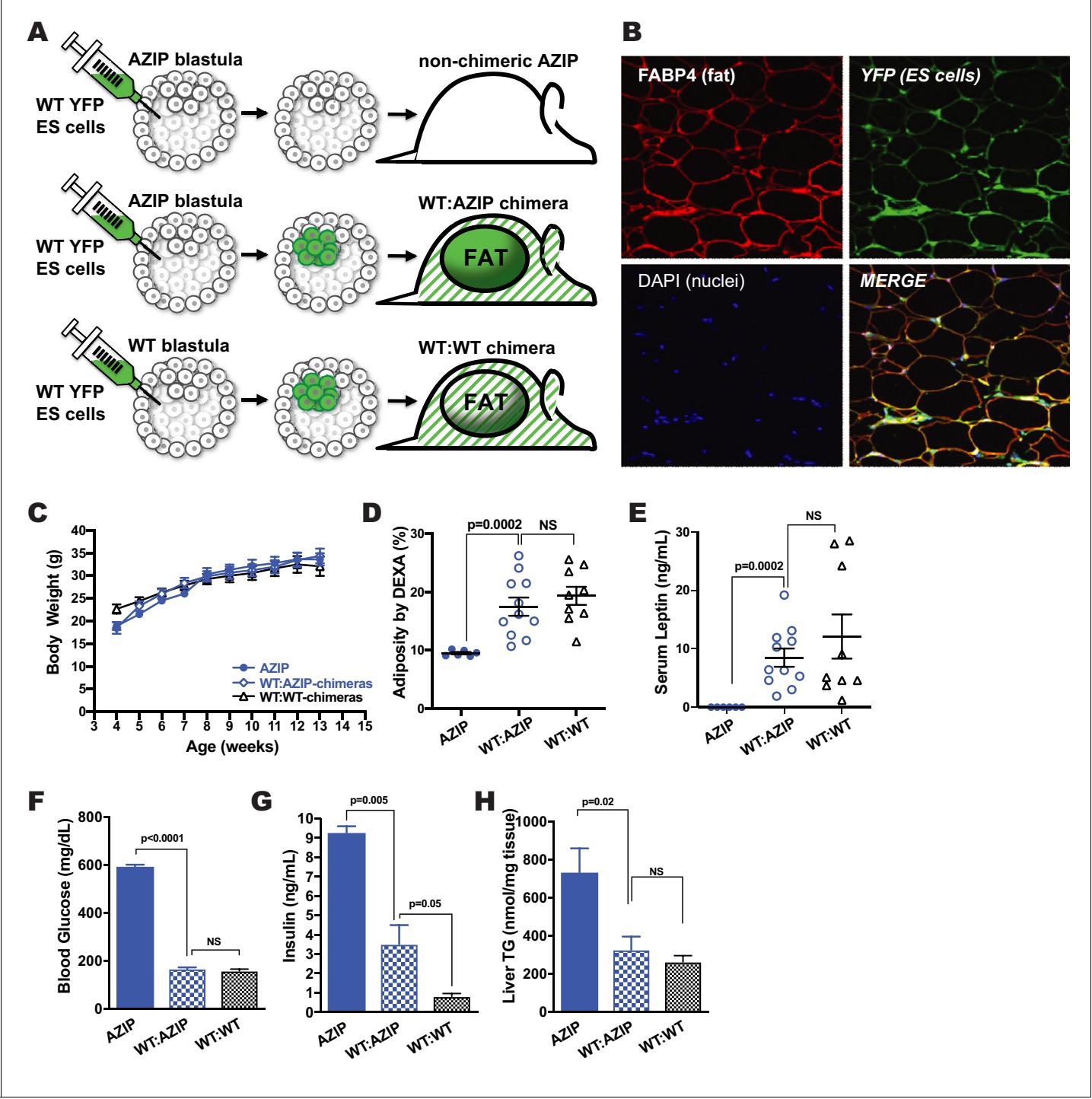

**Figure 1.** AZIP-Blastocyst Complementation with Pluripotent CAG-hTubYFP-ES Cells Using Microinjection. (**A**) Schematic representation of the blastocyst complementation procedure. AZIP- and WT-blastocysts isolated from donor females cannot be distinguished and are both injected with ES cells. ES-cells of B6-Tyr$^c$ background are labeled with CAG-driven human-tubulin-YFP (CAG-hTubYFP). Chimeric animals carrying both AZIP-cells and YFP-ES cells (WT:AZIP-chimeras) have normal amounts of adipose tissue exclusively derived from ES-progenies. Chimeric animals derived from injected WT-blastocysts are used as controls (WT:WT-chimeras) and have adipose tissue derived from both blastocyst- and ES-progenies. AZIP-blastocysts that failed to incorporate ES-cells remain lipodystrophic (non-chimeric AZIP). (**B**) Immunohistochemistry of paraffin section of inguinal adipose tissue of one WT:AZIP-chimera. All adipocytes, as shown to be FABP4-positive cells (red), are YFP-positive (green). Nuclei are stained with DAPI (blue). WT:AZIP-Chimeras in FVB Background Showed Normalized Adiposity and Mixed Metabolic Parameters. (**C**) Growth curve of AZIP (solid blue circle), WT:AZIP-chimera (open blue circle), and WT:WT-chimera (open black triangle). There is no significant difference at all points measured, from 4 to 14 weeks old,

*Figure 1 continued on next page*

*Figure 1 continued*

among all groups of animals. (D) Body adiposity as measured by DEXA of animals. AZIP animals have generalized lipodystrophy and have no visible gross adipose tissue and measured about 9.5% adiposity (attributed by fatty livers). ES cell complementation significantly increased (p=0.0002) adiposity to wild-type levels, 17.5% for WT:AZIP-chimera and 19.3% for WT:WT-chimera. (E) Serum leptin measurements. AZIP mice showed no detectable level of leptin and complementation restored leptin to wild-type level, 8.5 ng/mL for WT:AZIP-chimera vs. 12.1 ng/mL for WT:WT-chimera. (F) Blood glucose levels at 12 weeks old. In FVB, AZIP animals are severely diabetic with >600 mg/dL blood glucose. Blood glucose level is normalized by complementation. (G) Insulin level at 12 weeks. Consistent with glucose level, WT:AZIP-chimeras have significantly lower insulin in serum, but it is not completely normalized to wild-type level. (H) Liver triglyceride level at 14 weeks old. As a result of adipose tissue being restored, WT:AZIP-chimeras have normalized liver triglyceride levels, which is significantly lower than in AZIP animals. From (A) to (E) n = 6 for AZIP (solid blue), n = 11 for WT:AZIP-chimera (patterned blue), and n = 9 for WT:WT-chimera (black). For (F), n = 3 for AZIP (solid blue), n = 5 for WT:AZIP-chimera (pattern blue), and n = 7 for WT:WT-chimera (black).

The online version of this article includes the following figure supplement(s) for figure 1:

**Figure supplement 1.** Body adiposity by DEXA of AZIPs, WT:AZIP- and WT:WT-chimeric animals vs % chimerism.

## ES-Cell complementation in Chow-Fed AZIP(FVB) Mice

Eleven WT:AZIP(FVB)-chimeras were generated by injecting WT B6 ES-cells into AZIP(FVB)-blasto-cysts. These animals were compared with the adult AZIP littermates that showed no chimerism (six animals) and exhibited the full spectrum of abnormalities associated with lipodystrophy (this group is referred to as non-chimeric AZIPs, or just AZIP) and with WT:WT-chimeras that received ES-cells but did not carry the *A-ZIP* transgene (nine animals). The three groups of animals were fed a chow diet and showed similar growth curves between 4 weeks to 14 weeks (*Figure 1C*). Adipose tissue mass was reduced in the non-chimeric AZIP animals but was equivalent in WT:WT- and WT:AZIP-chimeras; non-chimeric AZIP 9.5 ± 0.2%, WT:AZIP-chimeras 17.5 ± 1.6% and WT:WT-chimera 19.6 ± 1.6% (p=0.0002 for non-chimeric AZIP vs. WT:AZIP chimeras, see *Figure 1D*). The levels of adipose tissue mass among the WT:AZIP chimeras was independent of the percent chimerism and animals with either low or high levels of chimerism showed an equivalent adipose tissue mass. (*Figure 1—figure supplement 1*). This suggested that for AZIP FVB animals fed a chow diet, the extent of adiposity was not affected by the degree of chimerism. Consistent with this, while serum leptin was not detectable in non-chimeric AZIPs, serum leptin in WT:AZIP-chimeras (8.5 ± 1.6 ng/mL) was equivalent to the level in WT:WT chimeras (12.1 ± 3.8 ng/mL; p=0.0002, and no significance respectively) (*Figure 1E*).

At 12 weeks, the hyperglycemia of AZIP(FVB) mice was completely normalized in WT:AZIP chimeras: 591.8 ± 8.2 mg/dL in non-chimeric AZIP animals (upper detection level of glucometer is 600 mg/dL) vs. 164.9 ± 6.7 mg/dl in the WT:AZIP-chimeric mice (p<0.0001; *Figure 1F*). Liver triglyceride (liver TG) was normalized to WT levels in the WT:AZIP-chimeras with a reduction from 731 ± 127 mmol/mg tissue in non-chimeric AZIP to 326 ± 69 mmol/mg tissue in WT:AZIP-chimeric animals (p=0.02; *Figure 1H*). WT:AZIP-chimeric animals had a plasma insulin level of 3.5 ± 1.0 ng/mL, vs. 9.3 ± 0.1 ng/mL for non-chimeric AZIP animals (p=0.005). However the insulin levels of the WT:AZIP-chimera was still significantly higher than WT:WT-chimera at 0.8 ± 0.04 ng/mL (p=0.05; *Figure 1G*), raising the possibility that subtle differences in the degree of chimerism could influence adiposity and that smaller levels of adiposity could lead to insulin resistance.

These results show that ES-cells can reconstitute the adipose tissue mass in non-obese FVB AZIP animals on a chow diet and that the resulting adipocytes can function normally to suppress the metabolic abnormalities of lipodystrophy. These data also show that in chow fed FVB derived chimeric mice, even a very limited amount of chimerism appears to be sufficient to restore normal adipose tissue mass although the residual insulin resistance in the WT:AZIP-chimera indicated that this restoration may not be totally complete. We thus set out to repeat the studies in diet induced obese (DIO) animals. However, since C57BL/6J is the strain that is most prone to develop obesity on a high fat diet (HFD) (*Montgomery et al., 2013*), we transferred the *A-ZIP* transgene to the C57BL/6J strain by backcrossing the transgene for six or more generations. We next measured adipose tissue mass in C57BL/6J WT:AZIP-chimeric animals on a chow diet.

## Adipose tissue mass in chow fed A-ZIP chimeras on B6 background

Genetic background can have a significant impact on the phenotype of lipodystrophic *A-ZIP* transgenic mice. On the FVB background, lipodystrophic animals develop severe metabolic

abnormalities, including hyperglycemia, hyperinsulinemic, hyperlipidemia, and liver steatosis (*Moitra et al., 1998*) (also see above). In contrast, C57BL/6J A-ZIP (lipodystrophic) mice have been previously reported to show reduced adipose tissue mass, low leptin levels and very prominent hepatic steatosis with only mild insulin resistance and normoglycemia (*Haluzik et al., 2004*; *Colombo et al., 2003*). Consistent with previous reports, the C57BL/6J AZIP animals we generated showed reduced fat mass, mild insulin resistance, euglycemia, and hepatic steatosis (see below).

To study the effect of genetic background on adipose complementation, we compared the phenotype of 17 WT:AZIP(B6)-ES cell chimeric animals fed a standard chow diet to two groups of control animals: non-chimeric littermate AZIP animals that did not incorporate ES-cells (17 animals) and WT:WT(B6)-chimeras that had received injected ES cells (19 animals). At three weeks of age, non-chimeric B6 AZIP-transgenic mice showed a reduced body weight compared to the chimeric WT:AZIP and WT:WT animals (p<0.01) but this difference diminished in older animals (*Figure 2A*). There was also a small but significant difference in fat mass (when corrected for the very enlarged fatty liver of the AZIP mice) and serum leptin among the three groups with the WT:AZIP animals intermediate between the other two (*Figure 2—figure supplement 1A* and *Figure 2—figure supplement 1B*). However, despite their slightly lower adiposity relative to the WT:WT chimeras, there was no effect of the level of chimerism on adiposity or serum leptin in the WT:AZIP chimeras. (*Figure 2B*). Thus, similar to FVB animals on a chow diet, C57Bl/6J WT:AZIP animals with low levels of chimerism showed similar levels of adiposity and serum leptin to those with high levels of chimerism (*Figure 2B and C*). The correlation between serum leptin and adiposity was the same in WT:AZIP and WT:WT chimeras suggesting that leptin production by adipocytes was similar in WT:AZIP adipose tissue. Consistent with this, histologic analysis of the adipose tissue of WT:AZIP- and WT:WT-chimeras with comparable levels of chimerism showed similar fat cell sizes suggesting the decrease in adiposity in the WT:AZIP chimeric animals was a result of a decreased cell number (*Figure 2—figure supplement 1C*). The effect of chimerism on cell number was confirmed in subsequent experiments directly measuring this and adipocyte size in chimeric animals on a high fat diet, see below.

We next compared metabolic parameters including glucose, insulin and liver steatosis in the chimeric animals. C57BL/6J A-ZIP mice, while showing a small increase in plasma glucose, are less hyperglycemic than AZIP(FVB) mice (see *Figures 1F* and *2E*). Plasma glucose was significantly reduced in WT:AZIP and WT:WT chimeras, compared to non-chimeric AZIPs (*Figure 2E*) but the reduction of glucose did not correlate with the level of chimerism (*Figure 2—figure supplement 1D*). C57BL/6J AZIP mice are hyperinsulinemic and the WT:AZIP-chimeras had significantly lower plasma insulin levels compared to non-chimeric AZIP with reduced insulin levels that were equivalent to those of WT:WT-chimeras; controls (117 ± 10 ng/mL in AZIPS vs. 8.2 ± 3.6 ng/mL in WT:AZIPs, p<0.0001; *Figure 2F*). Similarly, at 24 weeks, the enlarged liver associated of C57Bl/6J AZIPs was fully normalized in the WT:AZIP-chimeras and the liver size was indistinguishable from the size in the WT:WT-chimeras (5.8 ± 0.5 for AZIPs vs 2.1 ± 0.2 g in WT:AZIPs; p=6xE-4 and 1.7 ± 0.1 g for WT:WTs; *Figure 2G*). Here again there was no significant correlation between the level of chimerism and liver weight (*Figure 2—figure supplement 1E*).

In aggregate, these data show that even low amounts of chimerism can correct the features of the metabolic syndrome of B6 AZIP animals on a chow diet. However, similar to the FVB chimera, the adiposity and plasma leptin concentration of the WT:AZIP chimeras was slightly lower than in the WT:WTs chimeras raising the possibility that low chimerism, potentially indicating a lower number of adipocyte progenitors, may be constraining the ultimate size of the adipose tissue mass. To shed further light on whether the number of adipocytes can be limiting, we next placed the C57Bl/6 chimeric animals on a high fat diet which in this strain leads to the development of severe obesity.

## Adipose tissue mass is correlated with chimerism in C57BL/6J WT:AZIP-Chimeras On a High Fat Diet

A total of 75 male B6 A-ZIP chimeric animals were generated as previously described. 30 chimeras were WT:WT and 45 chimeras had the WT:AZIP genotype. Of these 45 animals, a total of 24 animals had less than one percent chimerism and are referred to as AZIP (or non-chimeric AZIPs) while 21 animals had had at least some degree of chimerism. All animals were placed on a high fat diet (HFD 60% calorie from fat) beginning at 6 weeks of age and growth curves were analyzed serially between six and 24 weeks of age when the animals were sacrificed (*Figure 3A*). We found greater variability of weight among the animals on a HFD compared to those on a chow diet, and as early as week 11

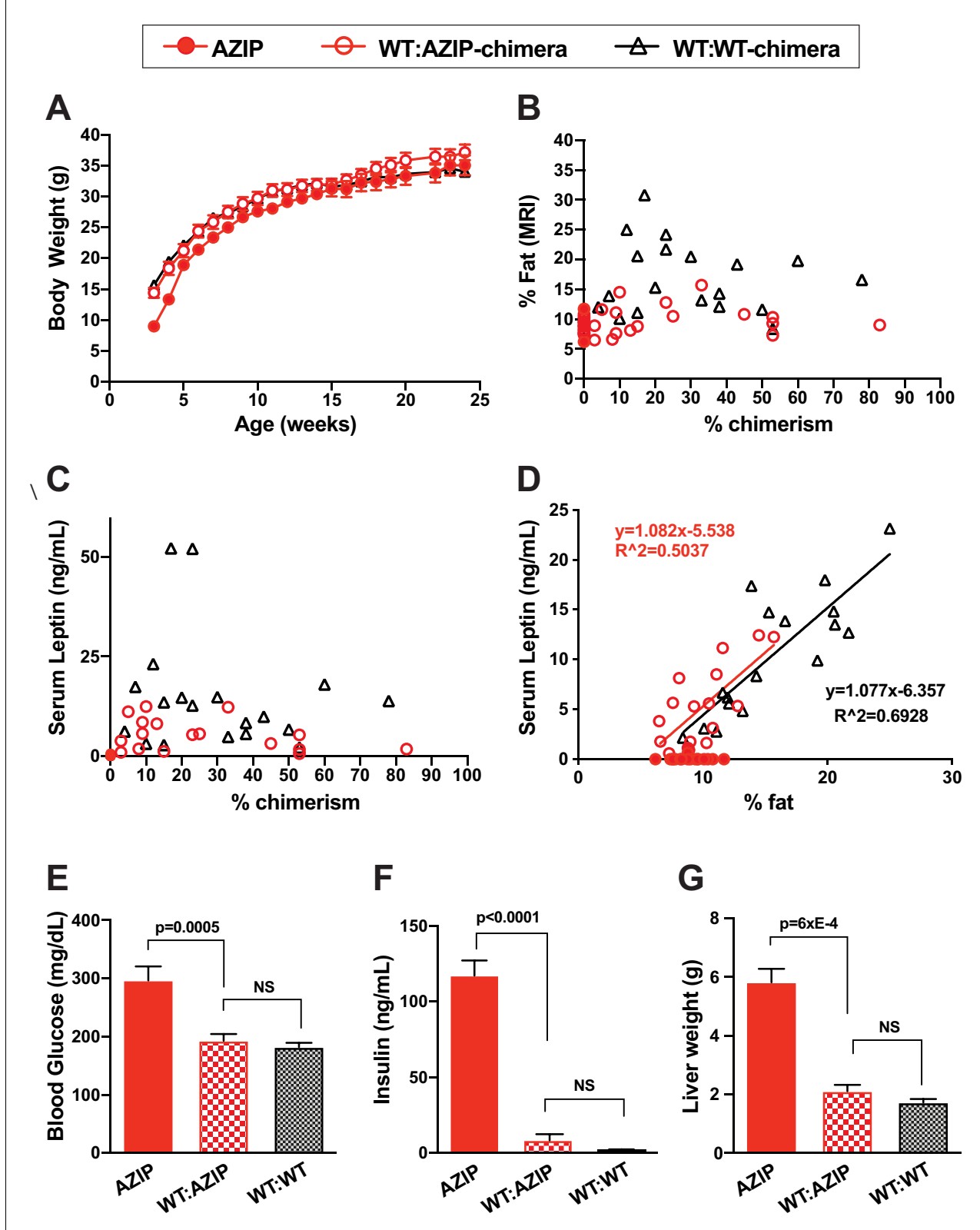

**Figure 2.** WT:AZIP-Chimeric mice in b6 background have normalized adiposity and fully complemented metabolic parameters on chow diet. (**A**) Body weight of AZIPs (solid red), WT:AZIP-chimeras (open red), and WT:WT-chimeras (black triangle) from 3 weeks to 24 weeks old. No difference in body weight was observed between WT:AZIP- and WT:WT-chimeras at all ages measured, except for AZIP mice that show significantly lowered body weight at weaning age (3 and 4 weeks old) but no difference is detected afterwards. (**B**) Body fat content measured by MRI at 12 weeks versus chimerism.

*Figure 2 continued on next page*

Figure 2 continued

There were no significant differences between AZIP and WT:AZIP-chimeric mice but WT:AZIP-chimeric mice are generally leaner than WT:WT-chimeric mice. (C) Serum leptin levels versus chimerism. AZIP mice have non-detectable levels of leptin and WT:AZIP-chimeras show serum leptin levels ranging from 0.6 to 12.5 ng/mL that is intermediate to AZIPs and WT:WT chimeras. No correlation is seen between chimerism and leptin levels in WT:AZIPs and WT:WTs. (D) Serum leptin and percent fat content correlate well in both WT:AZIP- and WT:WT-chimeras. There is no significant difference between the slopes of the two linear regressions. Two statistical outliers of WT-chimera with high leptin were excluded. Together, the results show that AZIP-chimeras form adipose tissue with normal leptin production despite a general reduction in adiposity. (E) Blood glucose levels at 12 weeks old. AZIPs are significantly higher than WT:AZIP and WT:WT-chimeras (297 ± 24 mg/dL for AZIPs vs. 193 ± 11 mg/dL for WT:AZIPs p=0.0005) and WT:AZIPs and WT:WT were the same (193 ± 11 mg/dL for WT:AZIPs vs 182 ± 7 mg/dL for WT:WTs, p=0.4). (F) Insulin level at 12 weeks. AZIP hyperinsulinemia was rescued in WT:AZIP-chimeras, from 117 ± 10 ng/mL to 8.2 ± 3.6 ng/mL (p<0.0001). (G) Liver weight. Liver weight is normalized by complementation (5.8 ± 0.5 for AZIPs vs 2.1 ± 0.2 g in WT:AZIPs; p=6xE-4 and 1.7 ± 0.1 g for WT:WTs). For (A) to (F), n = 17 for AZIP, n = 17 for WT:AZIP, and n = 19 for WT:WT-chimeras. For (G), n = 9 for AZIP, n = 12 for WT:AZIP and n = 14 for WT-chimeras, and liver analysis are performed at 6 months old.

The online version of this article includes the following figure supplement(s) for figure 2:

Figure supplement 1. B6 Chimeras on Chow Diet.

(after five weeks of HFD) we found that the WT:AZIP-chimeras as a group had significantly lower weights relative to the WT:WT-chimeras and significantly higher weights than the non-chimeric AZIPs. At the conclusion of the study (24 weeks old and after 18 weeks of HFD), WT:WT chimeras weighed an average of 56.6 ± 1 grams while non-chimeric AZIP animals only weighed an average of 38.1 ± 0.8 grams. WT:AZIPs with very low chimerism (1–10%) weighed 36.3 ± 0.6 grams, WT:AZIPs with medium chimerism (10–30%) weighed an average of 40.6 ± 1.2 grams and WT:AZIPs with high chimerism (>30%) weighed 44.6 ± 1.8 grams (*Figure 3A*). When the level of adiposity was analyzed relative to percent chimerism, a clear pattern emerged in which weight-gain correlated with the degree of chimerism with WT:AZIP animals with the highest percentage chimerism gaining significantly more weight than chimeras with a lower level of chimerism (*Figure 3B*; linear regression of WT: AZIPs is Y = 0.1876*X+36.00, $R^2$ = 0.4721). As expected, weight gain among WT: WT chimera was independent of chimerism (*Figure 3B*; WT:WT linear regression is Y = −0.04248*X+57.26, $R^2$ = 0.02877).

On a HFD, the percent fat of WT:AZIPs, assessed using MRI, was intermediate between that of non-chimeric AZIPs and WT:WT-chimera and, as was the case for weight, correlated with the extent of chimerism. (*Figure 3C*). Consistent with the previous studies of the chow fed animals, WT:AZIP-chimeras were markedly less obese compared to WT:WT chimeric animals. The increase in fat percentage correlated linearly with an increase in chimerism such that at the highest levels of chimerism, adiposity approached that of the WT:WT chimeric animals (*Figure 3C*; Y = 0.3128*X + 13.28, $R^2$ = 0.5633 for WT:AZIPS on HFD and Y = 0.01217*X + 42.40, $R^2$ = 0.003468 for WT:WTs on HFD). While the non-chimeric AZIPs did show approximately 10% body fat, as mentioned, this low level of adiposity was fully accounted for by the enlarged, steatotic liver in these animals. Thus, when the liver was removed post-mortem, the body fat percentage of non-chimeric AZIPs fell to 1.2% ± 0.1 (Data not shown). Consistent with the measures of adiposity, the serum leptin of WT:AZIPs was also intermediate between that of AZIPs and WT:WTs and, similar to adiposity, leptin levels increased linearly with increasing chimerism in WT:AZIPs (*Figure 3D*; linear regression of WT:AZIPs Y = 0.6495*X-0.1076, $R^2$ = 0.09426). Leptin levels were unaffected by the extent of chimerism in WT:WT chimera (Y = 0.2372*X+112.7, $R^2$ = 0.05183) while the non-chimeric AZIP animals had undetectable plasma leptin levels. These data show that when WT:AZIP-chimeras are placed on a high fat diet, lower levels of chimerism limits weight gain and adiposity, as well as the amount of leptin that circulates. Importantly, the WT:AZIP-chimera with ~80% chimerism, or above, became markedly obese with weights that approached that of wild-type animals. This indicates that the injected ES cells were capable of reconstituting the adipose tissue mass but only when the level of chimerism is high (*Figure 3B*). We next looked at the metabolic phenotype associated with the different levels of chimerism to see if decreased adiposity in the animals with low levels of chimerism negatively impacted metabolic state in DIO animals.

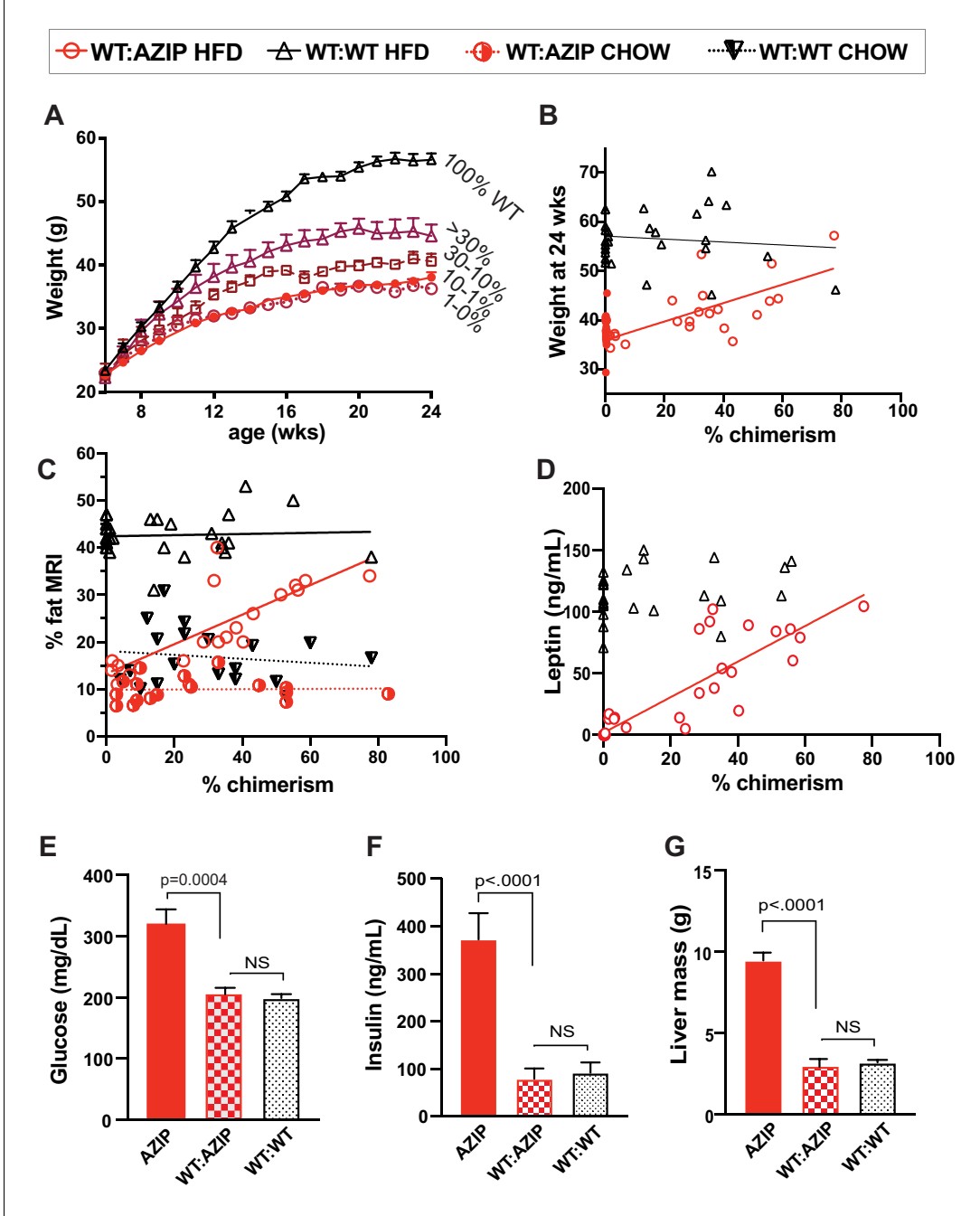

**Figure 3.** In high-fat-diet animals, adiposity and leptin are only partially complemented by chimerism while several metabolic indicators show complete rescue of the AZIP phenotype. (**A**) Body weight of five groups of chimeric animals on high fat-diet between 6 and 24 weeks of age. Final average weights are AZIPs (0–1% chimerism) 38.1 ± 0.8 g (closed red dot, n = 23), 1–10% WT:AZIP at 36.3 ± 0.6 g (open red circle. n = 5), 10–30% WT:AZIP 40.6 ± 1.2 g (open red square, n = 4),>30% WT:AZIP 44.6 ± 1.8 g (open red triangle, n = 12) and WT:WT with 56.6 ± 1.0 g (black triangle, n = 30). (**B**) Weight at 24 weeks (after 18 weeks of HFD) vs. chimerism. Linear regression of the WT:AZIP (red circle) weights is Y = 0.1876*X+36.00, R2 = 0.4721 and that of WT:WT (black triangle) is Y = −0.04248*X+57.26, R2 = 0.028). (**C**) Percent body fat measured with MRI versus percent chimerism. The linear regression of WT:AZIP chimeras on chow (red half-closed circle; from **B**) and HFD (red open circle) are Y = 0.003060*X+9.885, R2 = 0.0008 and Y = 0.3128*X+13.28, R2 = 0.56, respectively. WT:WT chimeras on chow (black inverted triangle, from **C**) and HFD (open black triangle) are Y = −0.04184*X+18.12, R2 = 0.01981 and Y = 0.01217*X+42.40, R2 = 0.003468, respectively. (**D**) Leptin levels vs. chimerism. Consistent with increasing fat, leptin increases with increasing % chimerism in WT:AZIP animals (Y = 1.443*X+1.758; R2 = 0.78), but not in WT:WT chimeras (Y = 0.2372*X+112.7, R2 = 0.05). (**E**) Ad libitum glucose levels at 24 weeks in three groups of animals. Blood glucose is corrected by complementation (WT:WT 200 ± 6.4 mg/dl, WT:AZIPs 207 ± 14, AZIP 320 ± 24 mg/dl). (**F**) Insulin at 24 weeks in three groups of animals. Insulin level is normalized by wild-type complementation in AZIP animals (AZIPs 373 ± 55 ng/mL, WT:AZIPs 79 ± 22 ng/mL, WT:WT 92 ± 21 ng/mL). (**G**) Liver mass at 24 weeks in three groups of animals on HFD. *Figure 3 continued on next page*

Figure 3 continued

Liver size of AZIPs (9.5 ± 0.4 grams) is normalized in all chimeric WT:AZIP animals (3.1 ± 0.2 grams) to WT:WT size (3.2 ± 0.2 grams). In (A) to (C) and (E) to (G), AZIPs n = 24, WT:AZIPs n = 21, and WT:WT n = 30. In (D), AZIPs n = 24, WT:AZIPs n = 21, and WT:WT n = 29. All readings done at 24 weeks after 18 weeks of HFD.

The online version of this article includes the following figure supplement(s) for figure 3:

**Figure supplement 1.** B6 Chimeras on High Fat Diet.

## Metabolic effects of reduced adipose tissue mass in C57BL/6J A-ZIP Chimeric Animals Fed a High Fat Diet

Recent studies have indicated that a subset of hyperinsulinemic patients have reduced adipose tissue mass, suggesting the possibility that a reduced storage capacity of the adipose depot can have metabolic consequences (*Scott et al., 2014*; *Yaghootkar et al., 2014*). We thus assessed whether a decreased fat storage in adipose tissue can have similar consequences in mice by comparing the metabolic phenotype of the WT:AZIP- and WT:WT- chimeras. We found that, as expected, non-chimeric AZIPs (chimerism below one percent) displayed severe metabolic consequences including hyperglycemia, hyperinsulinemia and hepatomegaly (*Figure 3E, F and G*, *Figure 3—figure supplement 1A, B, C and D*). However, despite the fact that there was a reduced level of adiposity, all chimeric HFD WT:AZIP animals, regardless of degree of chimerism, showed similar levels of ad libitum glucose, fasting glucose, fasting insulin levels and liver mass relative to the WT:WT chimeras (*Figure 3D, E and F*, *Figure 3—figure supplemnet 1D and F*). Thus restoring even a very small level of adipose tissue in chimeric animals was capable of normalizing glucose and insulin to the levels seen in the WT:WT chimeras. With levels of chimerism even as low as one percent, liver mass decreased from an average of 9.7 ± 0.5 to 3.1 ± 0.6 grams in WT:AZIPs with all WT:WT chimeras averaging 3.2 ± 0.2 grams of liver (*Figure 3G*). All animals showed similar levels of lean body mass (*Figure 3—figure supplement 1E*). Consistent with the above, WT:AZIPs with both high and low levels of chimerism all displayed the same levels of blood triglycerides and ketone bodies as wild-type chimeras (*Figure 3—figure supplement 1G, H, I and J*). Animals with low levels of chimerism also showed increased leptin levels relative to non-chimeric animals raising the possibility that even a small increase of leptin levels could improve and often normalize the metabolic phenotype of this subgroup (*Figure 3D*).

## Effect of chimerism on subcutaneous versus visceral fat accumulation among DIO WT:AZIP C57Bl/6J Chimeras

In these studies, we dissected and weighed both the subcutaneous inguinal (SubQ) and epididymal gonadal (visceral) fat pads and measured the individual mass of each depot (*Figure 4A*; blue shows SubQ and green denotes visceral, also, see material and methods). All WT:WT animals showed a similar weight of fat pads independent of the degree of chimerism. However, the WT:AZIP animals on a high fat diet showed distinct differences in the SubQ vs. visceral fat depots. SubQ fat pad mass correlated positively and linearly with percent chimerism and SubQ fat in the animals with the highest chimerism approached that of the SubQ fat mass in WT:WT animals (*Figure 4B*). In contrast, there was no correlation between chimerism and the size of the visceral fat pads. Visceral fat pads weighed a similar amount at all levels of chimerism, even at levels as low as one percent (*Figure 4C*). All WT:AZIP chimeras had somewhat lower visceral fat pad mass than in WT:WT chimeras. These results show that the subcutaneous and visceral depots are regulated differently and that degree of chimerism only affects the size of subcutaneous depot.

## Adipocyte cell size vs. number in DIO B6 chimeras

We calculated the average size of adipocytes in WT:WT and WT:AZIP chimeras using Adiposoft software to analyze H and E stained fixed sections. Adipocyte size and number were analyzed for both SubQ and visceral fat depots in animals with low and high degrees of chimerism (*Figure 4D*). There was no significant difference between fat cell size in either type of fat nor was there any correlation between size and the degree of chimerism (*Figure 4F*). Following from this, the number of adipocytes per SubQ fat pad in WT:AZIP chimeras significantly increased with increasing percent chimerism (*Figure 4E*). The average number of cells making up the WT:AZIP SubQ fat pad was 1.2 ± 0.2

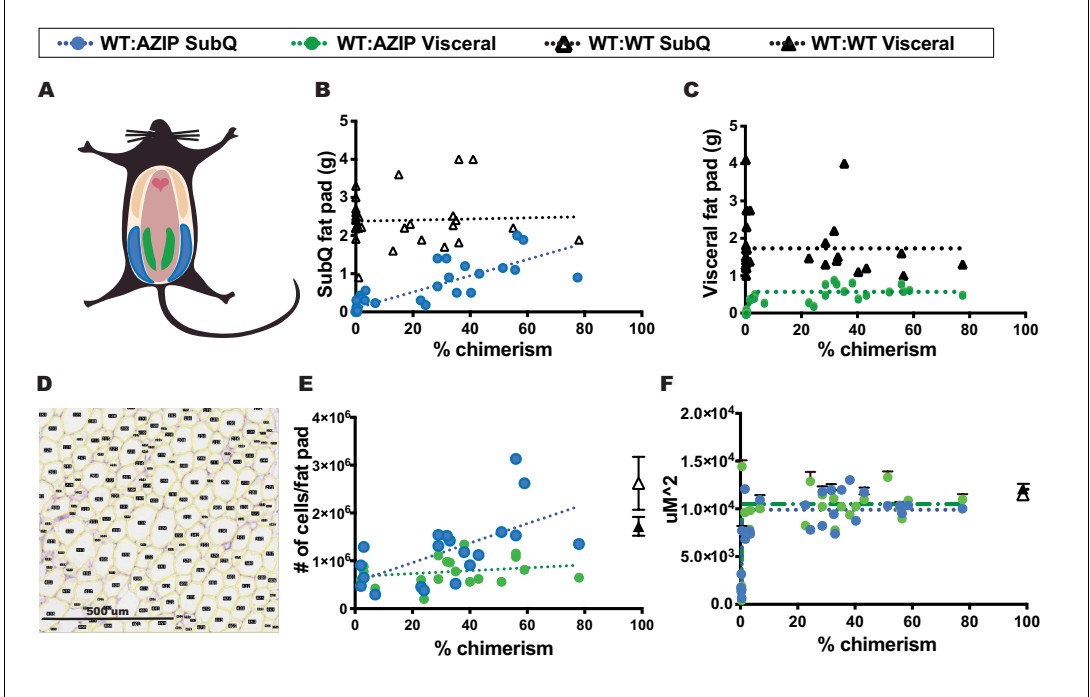

**Figure 4.** In high-fat-diet animals, the size of subcutaneous and visceral fats stores are differentially controlled. (A) Schematic representation of subcutaneous and visceral fat pads. Subcutaneous fat pads, shown in blue, and visceral fat pads, shown in green. (B) Subcutaneous fat pad mass vs. chimerism. Subcutaneous fat mass increases with increasing chimerism in WT:AZIP-chimeras (blue circles; linear regression Y = 0.02134*X+0.09579, R2 = 0.7147) but not in WT:WT-chimeras (black triangles; linear regression Y = 0.001482*X+2.377, R2 = 0.002). (C) Visceral fat pad mass vs. chimerism. Visceral fat pad mass is independent of chimerism in both WT:AZIP-chimeras (green circles) and in WT:WT-chimeras (black triangles). (D) Images of 5 uM fixed H and E stained adipocytes run through Fiji Adiposoft for individual size determination of adipocytes. Scale shown. (E) Number of cells per fat pad vs. chimerism. Number of subcutaneous adipocytes per fat pad increased with increasing chimerism (blue circles; 1/slope of the linear regression is 4.900e-005 with an R2 = 0.3737; Y = 20361*X+555585) whereas visceral adipocytes do not (green circles; 1/slope of the linear regression is 3.2e-4, R2 = 0.05135; Y = 3043*X+671862). WT:WT subcutaneous shown as open black triangle and WT:WT visceral adipocytes as solid black triangle. (F) Average area of top 20 adipocytes per 1,875,000 uM2 (1500 uM x 1250 uM) image. The size of WT:AZIP adipocytes, both subcutaneous (blue circles) and visceral (green circles) is independent of chimerism. Same results were found for WT:WT chimera adipocytes (black open triangle for subcutaneous adipocytes, black solid triangle for visceral adipocytes). In (A) to (F), AZIPs n = 24, WT:AZIPs n = 21, and WT:WT n = 30. Data collected at 24 weeks after 18 weeks of HFD.

E6 versus 2.6 ± 0.5 E6 in a WT:WT SubQ fat pad. The slope of the linear regression of the graph of SubQ cell number versus chimerism was significant for WT:AZIPs (1/slope = 4.900e-005 with an R2 = 0.3737; Y = 20361*X+555585). Conversely, in visceral fat depots, the number of adipocytes in the depots was unaffected by degree of chimerism. For visceral fat pads, there was an average number of 0.77 ± 0.6 E6 cells in WT:AZIPs and 1.7 ± 0.2 E6 cells in WT:WTs. The slope of the linear regression of the visceral fat WT:AZIP cell number versus chimerism was insignificant (1/slope = 3.2e-4, R2 = 0.05135; Y = 3043*X+671862; *Figure 4E*). These data indicated that there are reduced numbers of adipocytes in the subcutaneous depots of animals with low degrees of chimerism. Thus the differences in adiposity among chimeric animals resulted from differences in adipocyte number and not cell size, and there was a difference in adipocyte number in SubQ but not visceral fat.

## Discussion

In this study we asked whether the number of adipocyte progenitors, as inferred from the extent of chimerism in WT:AZIP-chimera (i.e.; AZIP blastocysts injected with wild-type ES cells), could be limiting for the size of the adipose tissue mass. We found that C57Bl/6J WT:AZIP-chimera fed a high fat diet with lower levels of chimerism developed a reduced adipose tissue mass compared to animals with a higher level of chimerism. We also found that differences in the size of the subcutaneous

depots in animals with lower levels of chimerism were fully accounted for by changes in cell number and that the percent chimerism did not affect adipocyte size in either visceral or subcutaneous fat pads. In contrast, diet (chow versus HFD) did affect cell size as adipocytes from animals on a chow diet were on average smaller than adipocytes from animals on a high fat diet. The fact that the level of chimerism was correlated with cell number in the subcutaneous adipose depots is consistent with the possibility that the level of chimerism reflects the number of progenitors. Thus, these results suggest that a reduced number of fat cell progenitors can limit the development of obesity. While studies directly assessing the number of progenitors would be necessary to further confirm this, suitable markers specific for adipocyte progenitors in vivo are not yet available.

Previous studies have differed with respect to whether fat cell progenitors can be limiting and restrict the accretion of adipose tissue as obesity develops. Thus while some studies have shown continued renewal of adipocytes in humans as well as new fat cell formation in rodents as obesity develops, other studies have suggested that there is a limited number of fat progenitors as animals enter maturity (*Spalding et al., 2008*; *Wang et al., 2013*; *Rivera-Gonzalez et al., 2016*). In a previous study on pancreas and liver development, the same approach revealed that while the size of the pancreas was lower in animals with lower levels of chimerism, the size of liver was not (*Stanger et al., 2007*). This result correlated well with the fact that liver possess considerable plasticity in size in response to various extrinsic signals such as after a partial hepatectomy (*Kang et al., 2012*). We were surprised to find that adipose tissue was more similar to the pancreas rather than liver. Owing to the massive increase of body weight that can be observed in adult animals placed on a HFD or after hypothalamic lesions (*Leibowitz et al., 1981*), we had anticipated that the extent of wild-type complementation (chimerism) would not be limiting, (i.e.; analogous to the liver). However, we found that this is not the case.

Our initial studies used lipodystrophic animals in which the A-ZIP transgene was bred on the FVB background because these mice were already available. A-ZIP mice express a truncated dominant negative form of Cebpα, a Jun family member, under the Fabp4 promoter. This Cebpα construct can bind to DNA but does not have an activation domain and the transgenic animals develop a severe lipodystrophy, similar to that in patients with generalized lipodystrophy, with little detectable adipose tissue and severe metabolic abnormalities. In these initial studies, we found that WT ES cells fully complemented the AZIP phenotype in FVB animals by restoring adiposity, serum leptin, blood glucose and liver triglycerides to wild-type levels. We did however note that the WT: AZIP-chimeric animals did have slightly elevated plasma insulin levels which was intermediate between that of non-chimeric AZIPs and WT:WT-chimeras. Insulin resistance is a hallmark of lipodystrophy and this result raised the possibility that there might be a subtle difference in the adipose tissue mass in the chimeric mice. We further tested this by inducing obesity in mice fed a high fat diet. However, because FVB mice do not develop the severe obesity of C57Bl/6J mice on a high fat diet, we transferred the transgene onto the C57BL/6J background (*Montgomery et al., 2013*). Consistent with a prior report, the metabolic phenotype of C57/Bl6J A-ZIP mice was different than that of FVB A-ZIP mice with milder diabetes and more severe hepatic steatosis. Here again, ES cell complementation of chow-fed C57Bl/6J WT:A-ZIP-chimeras led to a restoration of adipose tissue mass and the suppression of metabolic abnormalities independent of the level of chimerism. This provided an opportunity to test whether the ES cells could similarly restore adipose mass in Diet Induced Obese C57Bl/6J WT:AZIP-chimeras fed a high fat diet.

In DIO C57Bl/6J WT:AZIP-chimeras, there was a clear relationship between the level of chimerism and the level of obesity, and importantly, the ES cells were capable of fully restoring adipose tissue mass but only at the highest levels of chimerism. As mentioned, the differences in adipose tissue mass between the animals with low and high chimerism was a result of decreased numbers of adipocytes while the adipocyte size and level of leptin expression per adipocyte was unchanged. This suggests that the number of adipocyte progenitors can be limiting consistent with numerous prior reports showing that differences in the proliferative potential of adipocyte progenitor cells can influence the propensity to (or resistance to) obesity later in life (*Faust et al., 1976*; *Faust et al., 1977*; *Hirsch and Han, 1969*; *Johnson and Hirsch, 1972*; *Greenwood and Hirsch, 1974*; *Greenwood et al., 1979*).

While recent data have suggested that limited storage capacity of lipid (with the resulting deposition outside adipose tissue) can lead to metabolic abnormalities, we found, somewhat to our surprise, that plasma glucose and insulin levels and liver size were normalized even in the animals with

low levels of chimerism compared to A-ZIP controls. These findings are consistent with previous studies showing that a small increase in fat mass from wild-type but not ob/ob mice is capable of suppressing the lipodystrophic phenotype (*Rodeheffer et al., 2008*; *Gavrilova et al., 2000*; *Colombo et al., 2002*). While the animals with low chimerism had lower levels of adiposity, they did express increased levels of leptin, relative to controls, suggesting that even low levels, perhaps very low levels, of this hormone can be sufficient to suppress these metabolic abnormalities. This is also in agreement with another study showing that very low levels of leptin, too low to cause weight loss, can still correct hyperglycemia and insulin resistance in ob/ob mice (*Hedbacker et al., 2010*). Recent studies have suggested that lean Type two diabetics have an oligogenic form of lipodystrophy and these data further suggest that relatively decreased leptin levels in this cohort may be an important component of their phenotype, a possibility that has not been formally tested (*Lotta et al., 2017*).

In the studies of DIO chimeric animals, fewer ES cells were injected so that a greater number of animals with low chimerism could be used to determine whether this would be limiting. In the studies of animals fed a chow diet, a total of 15 ES cells were injected into blastocysts of 15–20 cells. In the DIO animals, five ES cells were injected into blastocysts of the same size (in an attempt to generate low levels of chimerism). It is estimated that at least 80% of injected cells survive if the injection is successful and chimerism develops. Extensive data from the Rockefeller University core has shown that for CY 2.4 ES cells from the same clonal origin, the percentage of chimerism, assessed by coat color, is lower when the number of injected cells is reduced from 15 to five (Yang, Chingwen, personal communication Feb. 3, 2020). In addition to the measures of adipocyte number, data showing that ES cells, and progenitor cells derived from them, survive and proliferate similarly to the resident blastomeres suggest that the extent of chimerism is likely to reflect the number of progenitors (*Eckardt et al., 2011*).

In these studies, we also noted clear differences between subcutaneous and visceral fat depots. Numerous papers have also shown that these different fat depots differ with respect to their developmental origins, function anatomic location (*Klyde and Hirsch, 1979*; *Cohen and Spiegelman, 2016*). For example, changes in subcutaneous fat stores are less well correlated with the development of metabolic disease than are changes in the size of visceral fat depots (*Merlotti et al., 2017*). In humans, subcutaneous fat stores have been associated with improved metabolic function, whereas visceral fat accumulation correlates with metabolic dysfunction (*Chusyd et al., 2016*). Subcutaneous and visceral fat stores also show differences in innervation by the sympathetic nervous system and respond differently to leptin treatment (*Wang et al., 2020*; *Chi et al., 2018*). Finally, there are multiple reports of differences in the response of these depots as adiposity develops or after surgical fat removal (*Faust et al., 1977*; *Johnson and Hirsch, 1972*). Our results reinforce the conclusion that subcutaneous and visceral depots are distinct by showing that the size of the visceral depot is not influenced by the degree of chimerism while the size of the subcutaneous depots is.

Subcutaneous and visceral fat also have different embryonic origins. Subcutaneous inguinal white adipose tissue arises from the lateral mesoderm germ layer, as do blood cells which were used in this study to determine degree of chimerism. Perigonadal white adipose tissue has a more heterogeneous origin from within the mesoderm germ layer with contributions from both paraxial and lateral mesoderm (*Luong et al., 2019*). Because blood cells and subcutaneous adipose tissue share a similar embryonic origin in the lateral mesoderm, we consider it likely the level of chimerism in blood cells accurately reflects chimerism in subcutaneous adipose tissue. This is less certain for gonadal visceral fat which derives from all parts of the mesoderm germ layer. We did not observe a difference in the size of visceral adipose depots amongst our chimeric animals suggesting that either visceral fat is not dependent on progenitor number or alternatively that chimerism in blood did not accurately reflect the extent of chimerism among the visceral fat precursors. However, the increased adiposity in mice we observed was the result of increased amounts of subcutaneous, not visceral fat and thus this possibility would not alter our central conclusion.

In summary, the present study made use of an ES cell complementation strategy to analyze the effect of differing levels of chimerism on adipose tissue mass and insulin sensitivity. The data are consistent with the possibility that the number of adipocyte progenitors constituting subcutaneous adipose tissue can be limiting for adiposity when animals are placed on a HFD. We also found that the size and cell number of subcutaneous adipose depots are dependent on the level of chimerism while visceral adipocytes are not. Finally, our data show that restoration of only a very limited amount of adipose tissue can markedly improve the metabolic defects of lipodystrophic mice.

Together with previous reports, this suggests that the development of metabolic abnormalities in lipodystrophic animals may to a large extent be a result of decreased leptin production (*Rodeheffer et al., 2008*; *Gavrilova et al., 2000*; *Colombo et al., 2002*) though it is possible that other adipocyte factors or metabolic functions of adipose tissue also contribute.

## Materials and methods

### Generation of chimeric mice

AZIP(FVB) chimeras were generated by injecting CAG-driven tubulin-YFP-ES-cells (B6-Tyr$^c$; Gene Targeting Resource Center at Rockefeller) into 3.5 day blastocysts from breeding of wild-type FVB females (Jackson FVB/NJ Stock #001800) to AZIP(FVB) transgenic males (FVB-Tg(AZIP/F)1Vsn/J Stock# 004100) (*Moitra et al., 1998*). Since both the blastocyst and the ES-cells have white coat color, AZIP(FVB) chimeras are completely albino. Degree of chimerism was determined by the percent of YFP-genotype cells in blood DNA samples. AZIP(B6) chimeras were generated by injection B6-Tyr$^c$ ES-cells into 3.5 day blastocysts from breeding of wild-type C57BL/6J females to AZIP(B6) transgenic males. AZIP(B6) transgenic mice were generated by backcrossing AZIP(FVB) male to wild-type C57BL/6J females for more than seven generations. Chimerism of AZIP(B6) chimeras were scored by coat color and averaged from two independent observers. AZIP transgenic mice are genotyped according to previous protocol with forward primer 5'-CTG TGC TGC AGA CCA CCA TGG and reverse primer 5'-CCG CGA GGT CGT CCA GCC TCA (*Moitra et al., 1998*). Due to cold sensitivity of neonates, breeding pairs consist of AZIP male and wild-type females are housed in non-ventilated cages and in elevated temperature (27–30° C). AZIP-pups were weaned around 4 weeks old, upon which the animals are transferred to standard housing condition. No special housing condition was implemented for chimeric mice.

### Animal experiments

Insulin (Alpco Mouse Insulin ELISA Cat#80-INSMS-E01) and plasma Leptin (R and D Systems Mouse/Rat Leptin Quantikine ELISA Kit Cat#MOB00B) were measured by ELISA according to the manufacturers' protocols using serum samples collected by retro-orbital bleeding using EDTA coated capillaries (Drummond Calibrated Micropipettes Glass Capillaries with EDTA 100 µl Cat#2-000-100-E). Blood glucose was measured by tail vein sampling using a Breeze2 glucometer (Bayer SKU: BREEZE2METER UPC: 301931440010). High fat diet treatments used Research Diets Cat #D12492 Rodent Diet With 60 kcal% Fat. Body fat content was measured by Dexa-scan or MRI using an Echo-MRI 100H by EchoMRI, LL, which provides measurements of total lean mass and fat mass. Percent body fat was then manually calculated by dividing fat mass over total mass. To determine the percent fat in mice with severe hepatic steatosis, the liver was removed post-mortem and them mice were put through the MRI again immediately following dissection. For triglyceride quantification in liver, fresh liver tissue with known weight (between 60 to 80 mg) was homogenized using a homogenizer (Polytron PT1200E by Kinematica IE) in 5% NP-40/water on ice, followed the assay protocol outlined in Triglyceride Quantification Kit (Abcam Cat# ab65336). Subcutaneous adipose tissue was dissected out postmortem from the subcutaneous posterior inguinal white fat depots. Both sides were dissected out and weighed individually on a scale. Visceral fat depots were dissected out postmortem from the two visceral perigonadal white adipose tissue pads and weighed individually on a scale. The weights reported reflect the weight of one fat pad. For all results, only male chimeric mice are used. Blood triglycerides and ketone-bodies were measured using Cayman Chemical Company Triglyceride colorimetric Assay Kits (Cat #10010303) and B-Hydroxybutyrate Colorimetric Assay Kits (Cat #700190), respectively.

### Statistical analysis

Error bars indicate standard error of the mean. Student ttest was used to perform pair wise comparison between groups. Outliers were detected by ROUT and noted in *Figure 1* legends.

### Calculation of percent chimerism

Blood from all chimeric animals as well as ES cells were pelleted, and DNA was extracted using a Qiagen mini prep kit. YFP primers and primers for a house keeping gene (FABP4 in *Figures 1* and

2 and RPL23 in *Figures 3* and *4*) and a 7500 Fast Real-Time PCR System form Applied Biosystems were used to compare the difference of the threshold cycle ($C_T$) of YFP with the $C_T$ of the house-keeping gene for each sample using the delta-delta-$C_T$ method and to extrapolate a standard curve we used the DNA prep from the YFP ES cells being 100% YFP (100% wild type). qPCR was done in triplicate and gDNA preps were repeated twice for WT:WT chimeras and three times for WT:AZIP chimeras.

## Cell size calculations

Adipocytes were dissected and fixed in 10% formalin for 48 hr and sectioned at 5 uM and stained for H and E by Histowiz, Inc Representative 1.875 mm̂two images were analyzed using Fiji Adiposoft 1.14 software and excluding edges to obtain measurements of the areas of each adipocyte. Each image was additionally manually screened to make sure they contained no broken adipocytes or other artifacts. The top 20 adipocyte readings were averaged.

# Acknowledgements

JMF acknowledges support from JPB foundation. The author would also like to thank the core facilities at Rockefeller University for technical support, Aidan Aug for discussions on the measurement of the size of adipocytes, Ruben Peraza for ES cell injections into blastocysts, and Inna Piscitello for editing and administrative support.

# Additional information

### Funding

| Funder | Author |
| --- | --- |
| JPB Foundation | Jeffrey M Friedman |

The funders had no role in study design, data collection and interpretation, or the decision to submit the work for publication.

### Author contributions

Kristina Hedbacker, Yi-Hsueh Lu, Data curation, Formal analysis, Validation, Investigation, Project administration; Olof Dallner, Methodology, Writing - review and editing; Zhiying Li, Conceptualization, Investigation; Gulya Fayzikhodjaeva, Chiayun Han, Methodology; Kıvanç Birsoy, Formal analysis, Investigation, Methodology; Chingwen Yang, Conceptualization, Methodology; Jeffrey M Friedman, Conceptualization, Resources, Software, Supervision, Funding acquisition, Methodology

### Author ORCIDs

Kristina Hedbacker (iD) https://orcid.org/0000-0003-0543-8733
Olof Dallner (iD) https://orcid.org/0000-0001-5614-1716
Zhiying Li (iD) http://orcid.org/0000-0002-5586-7782
Jeffrey M Friedman (iD) https://orcid.org/0000-0003-2152-0868

### Ethics

Animal experimentation: This study was performed in strict accordance with the recommendations in the Guide for the Care and Use of Laboratory Animals of the National Institutes of Health. All of the animals were handled according to approved institutional animal care and use committee (IACUC) protocol (15770) of the Rockefeller University.

### Decision letter and Author response

Decision letter https://doi.org/10.7554/eLife.53074.sa1
Author response https://doi.org/10.7554/eLife.53074.sa2

## Additional files

### Supplementary files

• Transparent reporting form

### Data availability

No large data sets were generated. Figures 1D, 1E, Figure 1—figure supplement 1, Figure 2B, 2C, 2D, Figure 2—figure supplement 1A, 1B, 1D, 1E, Figure 3B, 3C, 3D, Figure 3—figure supplement 1A, 1B, 1C, 1D, 1E, 1F, 1H, 1J, Figure 4B, 4C, 4E, 4F all contain raw data points.

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
