## [Decision Letter]

**Acceptance summary:**

Your paper addresses the important question of what factors contribute to the expansion of specific adipose tissue depots in obese animals. The data presented in this study suggest that the size of subcutaneous adipose depots in mice depend on the number of progenitor cells that are present at birth, while visceral adipose depots do not. This study also shows that relatively small amounts of adipose tissue are sufficient to protect mice from metabolic disease upon high fat feeding.

**Decision letter after peer review:**

Thank you for submitting your article "Regulation of Adipose Tissue Mass by the Number of Embryonic Progenitor Cells" for consideration by *eLife*. Your article has been reviewed by two peer reviewers, and the evaluation has been overseen by a Reviewing Editor and Mark McCarthy as the Senior Editor. The reviewers have opted to remain anonymous.

The reviewers have discussed the reviews with one another and the Reviewing Editor has drafted this decision to help you prepare a revised submission.

The reviewers felt that your use of blastocyst complementation to determine whether embryonic constraints on adipocyte development influence subsequent body weight, fat mass, and metabolic outcomes is an innovative approach. Your system utilizes a lipodystrophic transgenic strain into which wild-type ES cells are introduced at the blastocyst stage such that all fat cells are ES-derived. An advantage of the system is that the resulting chimeric animals exhibit varying degrees of chimerism, and hence varying degrees of rescue (of various phenotypes) can be related to relative contributions of wild type cells. The main findings come from studies in which the transgene has been bred onto a C57Bl6 background and the mice fed a high-fat diet. Here, some phenotypes are rescued (e.g. weight, subcutaneous fat, leptin levels) and others are not (e.g. glucose and insulin levels, liver mass, visceral fat).

There were some concerns raised in the review process, however, and we ask you to consider the following main points in preparing an appropriately revised manuscript:

1) A major concern is that the text repeatedly asserts that many of the observed outcomes reflect constraints imposed by the number of progenitor cells. However, adipocyte progenitor cell number is never measured in the study but is merely inferred based on variation in the degree of chimerism. Thus, while the system has demonstrated that there are developmentally-imposed constraints on adipose mass, the conclusion that this is due to limits on embryonic progenitor cells is a likely, but unproven, interpretation. A suggestion for dealing with this is to amend the text and section headings of the manuscript to eliminate phrasing that equates degree of chimerism with "number of progenitor cells" and use the Discussion to advance the argument that progenitor cells are limiting for adipose mass. (The title can remain as is, since this is the major conclusion).

2) Related to the above is the conclusions drawn from the differences in the effectiveness of fat reconstitution to chimerism between the visceral and subcutaneous depots. The percent chimerism was calculated from blood cells, which are derived from the lateral plate mesoderm. Subcutaneous fat is also derived from the lateral plate mesoderm, and scoring is well founded. However, visceral fat is not derived from the same developmental lineage. Thus your conclusions on this point may be due to the method used to score chimerism. The discussion of the visceral fat data should detail this caveat and the conclusion about the potential differences in adipose depot biology should be amended. Also, the factors affecting subcutaneous and visceral adipose depot mass differ between males and females – were both used?. This issue should be discussed. Also, the identity of the subcutaneous and visceral depots studied should be stated in the Materials and methods.

3) The DEXA data for the FVB WT-AZIP chimeras shows a clear reconstitution of fat mass over AZIP that is indistinguishable from the WT-WT chimeras, but the MRI percent fat from the WT-AZIP B6 chimeras seem to have much lower fat reconstitution on chow (Figure 3B). Is there significant reconstitution of fat mass on chow with the B6 WT-AZIP chimeras? If not, the difference between strains should be discussed.

---

## [Author Response]

There were some concerns raised in the review process, however, and we ask you to consider the following main points in preparing an appropriately revised manuscript:1) A major concern is that the text repeatedly asserts that many of the observed outcomes reflect constraints imposed by the number of progenitor cells. However, adipocyte progenitor cell number is never measured in the study but is merely inferred based on variation in the degree of chimerism. Thus, while the system has demonstrated that there are developmentally-imposed constraints on adipose mass, the conclusion that this is due to limits on embryonic progenitor cells is a likely, but unproven, interpretation. A suggestion for dealing with this is to amend the text and section headings of the manuscript to eliminate phrasing that equates degree of chimerism with "number of progenitor cells" and use the Discussion to advance the argument that progenitor cells are limiting for adipose mass. (The title can remain as is, since this is the major conclusion).

We agree with this comment and in the edited manuscript have avoided using the wording ‘number of adipocyte progenitors’ and instead now refer to “extent of chimerism”. We also address the difference between the extent of chimerism and the number of progenitors in the revised Discussion and as suggested discuss the possibility that differences in the extent of chimerism is correlated the number of progenitors. We also discuss what is known about the extent of survival of injected ES cells in a chimeric animal.

2) Related to the above is the conclusions drawn from the differences in the effectiveness of fat reconstitution to chimerism between the visceral and subcutaneous depots. The percent chimerism was calculated from blood cells, which are derived from the lateral plate mesoderm. Subcutaneous fat is also derived from the lateral plate mesoderm, and scoring is well founded. However, visceral fat is not derived from the same developmental lineage. Thus your conclusions on this point may be due to the method used to score chimerism. The discussion of the visceral fat data should detail this caveat and the conclusion about the potential differences in adipose depot biology should be amended. Also, the factors affecting subcutaneous and visceral adipose depot mass differ between males and females – were both used?. This issue should be discussed. Also, the identity of the subcutaneous and visceral depots studied should be stated in the Materials and methods.

We have added a new section to the Discussion addressing the heterogeneous origins of gonadal visceral adipose tissue being both paraxial and lateral mesoderm. We now point out that the method for determining chimerism using blood cells that arise from lateral mesoderm is a more accurate prediction of true chimerism for subcutaneous adipose tissue (than visceral fat), which arise from the same sub-germ layer.

We also note that, in contrast to the subcutaneous depot, the size of the visceral fat adipose tissues are independent of the level of chimerism. Finally, we have added a more detailed description to the Materials and methods section stating the exact methods used to identify and dissect the different adipose tissue depots.

3) The DEXA data for the FVB WT-AZIP chimeras shows a clear reconstitution of fat mass over AZIP that is indistinguishable from the WT-WT chimeras, but the MRI percent fat from the WT-AZIP B6 chimeras seem to have much lower fat reconstitution on chow (Figure 3B). Is there significant reconstitution of fat mass on chow with the B6 WT-AZIP chimeras? If not, the difference between strains should be discussed.

The reviewer is correct that wildtype complementation appears more complete for restoration of adiposity in FVB animals than B6 animals. These results are also consistent with those measuring leptin levels in which are lower in B6 than FVB animals (compare current Figure 1D, with Figure 2—figure supplement 1A and Figure 1E with Figure 2—figure supplement 1B). As per a publication referenced in the paper, FVB animals are less predisposed to obesity than are B6 animals. This suggested to us that the ES cells are less able to fully complement the greater adipose tissue mass of B6 animals relative to FVB. We then confirmed this by feeding B6 lipodystrophic animals a high fat diet to induce obesity and the associated metabolic abnormalities and confirmed that when chimerism is low in DIO B6 WT:AZIP chimeric animals the ES cells are unable to fully reconstitute the entire adipose tissue mass when obesity develops.